# Novel Genetic Lineages of *Rickettsia helvetica* Associated with *Ixodes apronophorus* and *Ixodes trianguliceps* Ticks

**DOI:** 10.3390/microorganisms11051215

**Published:** 2023-05-05

**Authors:** Yana Igolkina, Valeriy Yakimenko, Artem Tikunov, Tamara Epikhina, Aleksey Tancev, Nina Tikunova, Vera Rar

**Affiliations:** 1Institute of Chemical Biology and Fundamental Medicine SB RAS, Lavrentiev Avenue 8, 630090 Novosibirsk, Russia; igolkina@inbox.ru (Y.I.); tikunova@niboch.nsc.ru (N.T.); 2Omsk Research Institute of Natural Foci Infections, Mira Avenue 7, 644080 Omsk, Russia; yakimenko_vv@oniipi.org (V.Y.); mail@oniipi.org (A.T.)

**Keywords:** *Ixodes Rickettsia helvetica*, “*Candidatus* Rickettsia uralica”, “*Candidatus* Rickettsia tarasevichiae”, co-feeding, genetic lineages, phylogenetic analysis

## Abstract

*Ixodes apronophorus* is an insufficiently studied nidicolous tick species. For the first time, the prevalence and genetic diversity of *Rickettsia* spp. in *Ixodes apronophorus*, *Ixodes persulcatus*, and *Ixodes trianguliceps* ticks from their sympatric habitats in Western Siberia were investigated. *Rickettsia helvetica* was first identified in *I. apronophorus* with a prevalence exceeding 60%. “*Candidatus* Rickettsia tarasevichiae” dominated in *I. persulcatus*, whereas *I. trianguliceps* were infected with “*Candidatus* Rickettsia uralica”, *R. helvetica*, and “*Ca*. R. tarasevichiae”. For larvae collected from small mammals, a strong association was observed between tick species and rickettsiae species/sequence variants, indicating that co-feeding transmission in studied habitats is absent or its impact is insignificant. Phylogenetic analysis of all available *R. helvetica* sequences demonstrated the presence of four distinct genetic lineages. Most sequences from *I. apronophorus* belong to the unique lineage III, and single sequences cluster into the lineage I alongside sequences from European *I. ricinus* and Siberian *I. persulcatus*. *Rickettsia helvetica* sequences from *I. trianguliceps*, along with sequences from *I. persulcatus* from northwestern Russia, form lineage II. Other known *R. helvetica* sequences from *I. persulcatus* from the Far East group into the lineage IV. The obtained results demonstrated the high genetic variability of *R. helvetica*.

## 1. Introduction

*Ixodes* spp. (Ixodidae, Ixodida, Arachnida) ticks are recognized vectors of different bacterial agents; the most epidemiologically important of them are spirochetes from *Borrelia burgdorferi* sensu lato species complex and relapsing fever group, as well as intracellular bacteria from the Anaplasmataceae and Rickettsiaceae families [1,2,3,4]. Several *Ixodes* species are common in Western Siberia of Russia, namely *Ixodes persulcatus*, *Ixodes pavlovskyi*, *Ixodes trianguliceps*, *Ixodes apronophorus*, *Ixodes lividus*, and *Ixodes crenulatus*. Of them, only *I. persulcatus* and *I. pavlovskyi* bite humans [5,6].

In many locations, several *Ixodes* species occur simultaneously. Thus, in the south taiga subzone of Western Siberia, *I. persulcatus* often occurs in sympatry with *I. trianguliceps*; moreover, in some locations, three *Ixodes* species, *I. persulcatus*, *I. trianguliceps*, and *I. apronophorus*, can coexist together [7,8]. All these tick species have three-host developmental cycles. Small mammals are the main hosts for the preimaginal stages of *I. persulcatus*, whereas *I. persulcatus* adults feed mainly on middle-size and large mammals [5,9]. All stages of *I. trianguliceps* and *I. apronophorus* feed predominantly on small mammals. European water vole (*Arvicola amphibious*) is considered to be one of the main hosts for *I. apronophorus* [6,7,9].

*Ixodes persulcatus* and *I. trianguliceps* have been tested for the presence of different bacterial agents in a number of studies. Both tick species can be infected with *Anaplasma phagocytophilum* and *B. burgdorferi* s.l.; in addition, *Ehrlichia muris* and *Borrelia miyamotoi* have been found in *I. persulcatus* [10,11,12,13]. As for rickettsial agents, *I. persulcatus* was most frequently infected with “*Candidatus* Rickettsia tarasevichiae” (up to 90% of infected ticks), and in rare cases, with *Rickettsia helvetica*, *Rickettsia heilongjiangensis*, *Rickettsia raoultii*, and *Rickettsia sibirica* [11,14,15,16]. Previous studies have demonstrated that *R. helvetica*, “*Ca*. R. tarasevichiae”, and “*Candidatus* Rickettsia uralica” can be detected in *I. trianguliceps* [14,17].

In contrast to *I. persulcatus* and *I. trianguliceps*, *I. apronophorus* is insufficiently studied. This tick has been recorded in many European countries, Siberia, and Kazakhstan [6,7,9,18,19]. A recent genetic study of *I. apronophorus* has demonstrated that it belongs to *I. ricinus*-*I. persulcatus* species complex within the subgenus *Ixodes* [8]; *I. trianguliceps* is a member of the Exopalpiger subgenus [9]. Although *I. apronophorus* does not bite humans, it may transmit infectious agents to *I. persulcatus* through infected animals or during co-feeding on the same small mammals and, thus, participate in the same enzootic cycle. It has been recently shown for the first time that *I. apronophorus* can be infected with *Borrelia bavariensis* and “*Candidatus* Borrelia sibirica” [13]. However, the ability of *I. apronophorus* to be infected with any Anaplasmataceae and Rickettsiaceae agents has not been studied to date.

In this study, we first examined the presence and genetic variability of *Rickettsia* spp. in *I. apronophorus* ticks collected in *I. persulcatus*/*I. trianguliceps*/*I. apronophorus* sympatric areas. The species diversity and genetic variability of *Rickettsia* spp. identified in different *Ixodes* species from the same sympatric areas were compared.

## 2. Materials and Methods

### 2.1. Sampling

Ticks were collected in two sampling sites within the forest zone in Omsk province, Western Siberia, Russia. The first site (Om-Bo) was located at the boundary between aspen-birch forests and southern taiga in the Bolsheukov district (56°46′ N, 72°03′ E), and the second site (site Om-Zn) was situated in the southern taiga subzone in Znamenskiy district (57°23′ N, 73°40′ E) (Figure 1). All animal experiments were approved by the Animal Welfare Act at the Omsk Research Institute of Natural Foci Infections, according to the guidelines for experiments with laboratory animals (Supplement to the Order of the Russian Ministry of Health, no. 755, of 12 August 1977). Animal use and experimental procedures were approved by the Bioethical Committee of the Omsk Research Institute of Natural Foci Infections (Protocol No.1, 20 March 2013; Protocol No. 4, 17 February 2016).

Wild rodents were captured in the site Om-Bo in June 2016 and in the site Om-Zn from June–September 2014–2015. Voles and mice of the genera *Myodes*, *Microtus* and *Apodemus* were caught using live traps, and European water voles were captured by steel traps. The species of trapped animals were determined based on morphological features. The animals were examined for the presence of attached ticks (larvae, nymphs and adults), which were removed with forceps.

The species and stage of ticks collected from animals were preliminarily determined using a stereo microscope MC-800 (Micros, Hunnenbrunn/Gewerbezone, Austria), according to morphological keys [9,20]. The tick species of all *Ixodes* spp. ticks were additionally determined using the multiplex PCR assay by ITS2 fragments as described previously [8]. For a subset of specimens, the species identities were confirmed by sequencing of ITS2 and/or mitochondrial cox1 gene fragments as previously described [8].

Some engorged and nearly engorged larvae and nymphs were stored at 10–15 °C for 1–2 weeks and then transported to the laboratory and allowed to molt into nymphs or adults, respectively. Other ticks were placed in sealed plastic tubes, which were then stored in liquid nitrogen until species determination and DNA extraction.

Hereinafter, ticks molted in the laboratory are named “molted ticks”, whereas ticks examined without preliminary molting are named “non-molted ticks”.

### 2.2. Tick Metamorphosis

For successful molting, partially engorged larvae and nymphs were fed to repletion on laboratory white mice. Each engorged tick was placed individually in a glass tube and incubated in the dark at 100% relative humidity at 24–26 °C until completion of molting. Molted ticks were individually frozen four weeks after molting and stored at −70 °C until DNA extraction.

### 2.3. DNA Extraction

Frozen ticks were homogenized with the MagNA Lyser Instrument using the MagNa Lyser Green Beads (Roche Diagnostics, Basel, Switzerland). Total DNA was extracted from crushed individual ticks using the Proba NK kit (DNA-Technology, Moscow, Russia) according to the manufacturer’s protocol. To prevent cross-contamination, DNA/RNA extraction, amplification, and PCR product detection were carried out in separate rooms. Aerosol-free pipette tips were used at each stage.

### 2.4. Detection and Genotyping of Rickettsia spp.

To detect *Rickettsia* spp. DNA, nested PCR was performed for the *glt*A gene with primers glt1-glt4. For correct species determination in the case of probably mixed infection, additional nested reactions were performed independently using primers RT1 and RT2, specific to “*Ca*. R. tarasevichiae”, and primers RH1 and RH3, specific to spotted fever group rickettsiae (SFGR) (Table 1). The amplified *glt*A gene fragments from all specimens positive for SFGR and some specimens positive for “*Ca*. R. tarasevichiae” were sequenced. For a number of specimens, fragments of 16S rRNA, *omp*A, *omp*B, *sca*4, and *htr*A genes, as well as groESL operon and 23S-5S IGS, were additionally amplified using primers specified in Table 1 and sequenced.

### 2.5. Sequencing and Phylogenetic Analysis

The PCR products were purified using GFX Columns (Amersham Biosciences, Piscataway, NJ, USA). Sanger reactions were performed using the BigDye Terminator V. 3.1 Cycling Sequencing Kit (Applied Biosystems, Foster City, CA, USA) in standard conditions specified in the BigDye Terminator V. 3.1 Cycling Sequencing Kit User Guide. Sanger reaction products were purified using CentriSep spin columns (Princeton Separations, Freehold, NJ, USA) and visualized with a 3500 Genetic Analyzer (Applied Biosystems, Foster City, CA, USA). Sequence analysis was performed with BlastN (http://www.ncbi.nlm.nih.gov/BLAST, accessed on 3 April 2023) and BioEdit (http://www.mbio.ncsu.edu/BioEdit/bioedit.html, accessed on 3 April 2023). Phylogenetic trees were constructed using the Maximum likelihood (ML) method based on the Tamura-Nei model in MEGA 7.0 with 1000 bootstrap replicates [21].


microorganisms-11-01215-t001_Table 1Table 1Primers used for detection *Rickettsia* spp.Amplified LocusOrganismReactionPrimer NamePrimer Sequences 5′-3′T^#^ (°C)References
*16S rRNA*
*Rickettsia* spp.Primary16S1gacgggtgagtaacacgtggg56[22]gene

16S2gtcttttagggatttgctccac



Nested16S3gatggatgagcccgcgtcag60



16S4gcatctctgcgatccgcgac

*gltA* gene*Rickettsia* spp.Primaryglt1gattgctttacttacgaccc52[22]


glt2tgcatttctttccattgtgc



Nestedglt3tatagacggtgataaaggaatc53



glt4cagaactaccgatttctttaagc

*ompA* gene*Rickettsia* spp.
*Conventional*
Rr190.70patggcgaatatttctccaaaa55[23]fragment I

190-701gttccgttaatggcagcatct

*ompA* gene*Rickettsia* spp.PrimaryAfnw_1ggcacaaatactttaacattacc52[24]


190-6808cacgaactttcacactacc
[23]

NestedAfnw_3aagcctactcctaaagagaatg53[24]


Afnw_4cgacagtctctagtgccg


*Rickettsia* spp.PrimaryA1taacattacaagctggaggaagcc58[22]


A2ttcagagcctgaccaccgg



NestedA5caagtgctggtgatgttacta56



A6tagttacatttcctgcacctac



*R. helvetica*
PrimaryAfn1_helvgtaatactagcatcaccgaaatcc55This study


190-6808cacgaactttcacactacc
[23]

Nested190-5125gcggttactttagccaaagg54



Afnw_4cgacagtctctagtgccg
[24]*ompA* gene*“Ca*. R. tarasevichiae”
*Conventional*
190-5125gcggttactttagccaaagg56[23]fragment IV

190-6013m *gcatcttytgcgttgyattac

*ompB* gene*Rickettsia* spp.PrimaryM59 Fccgcagggttggtaactgc55[25]


120-1497m *cctatatcgccggtaattgtagc



NestedBR1gttactaatggatttattcaagt53[22]


BR2gcataaacttgtccagcgat


*Rickettsia* spp.PrimaryB2f_5taaacttgctgacggtacag56[24]


B2f_2cgattatgccgttatcgcttccaag



NestedB2f_3gtagcctaacaaatgctcaaac52



120-2399cttgtttgtttaatgttacggt
[25]

*R. helvetica*
PrimaryB2f_3gtagcctaacaaatgctcaaac52



B2f_2cgattatgccgttatcgcttccaag



NestedB2f_1helvcagtacaattcgctcacaacac55This study


120-2399cttgtttgtttaatgttacggt


*Rickettsia* spp.PrimaryB1atatgcaggtatcggtact56[22]


B2ccatataccgtaagctacat



NestedB3gcaggtatcggtactataaac56



B4aatttacgaaacgattacttccgg


*Rickettsia* spp.Primary120-3462ccacaggaactacaaccatt52[25]


120-4879m *tagaagtttacacggacttttagag



NestedB3f_3fgctggacctgaagctggagc55[24]


120-4879m *tagaagtttacacggacttttagag
[25]*sca4* gene*Rickettsia* spp.PrimaryD1fatgagtaaagacggtaacct52[26]


D1876rm *tagtttgttccgccgtaatc



Nestedsc1f_3gatgtaggtgatgaactctg52[24]


D1390rcttgcttttcagcaatatcac
[26]

Primarysc4-1atgtctctgaattaagcaatgc52[22]


Rj2837rcctgatactacccttacatc
[27]

Nestedsc4-5ccggcacaacaacaattgatg50[22]


sc4-6cctttaccagctcatctactt



Primarysc4-3aattattaggctctgtattaaaga52[22]


D3069rtcagcgttgtggaggggaag
[26]

Nestedsc4-5ccggcacaacaacaattgatg52[22]


sc4-7ctctcttttaataggtgttgatt
[24]*htrA* gene*Rickettsia* spp.
*Conventional*
17k-5gctttacaaaattctaaaaaccatata55[28]


17k-3tgtctatcaattcacaacttgcc

23S-5S IGS*Rickettsia* spp.
*Conventional*
RCK/23-5-Fgataggtcrgrtgtggaagca55[29]


RCK/23-5-Rtcgggaygggatcgtgtgtttc

groESL*Rickettsia* spp.PrimaryRic-ESL-F1ggtaaatgggcaggyaccgaa60[30]operon

Ric-ESL-R1gaagcaacrgaagcagcatctt



NestedRic-ESL-F2atcgttatgaaagaaagcgayg58



Ric-ESL-R2agwgcagtacgcactactttagc

m *—modified primers, T^#^—Annealing temperature.


### 2.6. Statistical Analysis

Differences in the prevalence of causative agents between tick species were computed with the Pearson χ^2^ goodness-of-fit test (http://www.socscistatistics.com/tests/chisquare/, accessed on 20 February 2023). *p* < 0.05 was regarded as statistically significant.

### 2.7. Nucleotide Sequence Accession Numbers

Nucleotide sequences determined in this study were deposited in the GenBank database under accession numbers: ON863706-ON863712; OQ092468-OQ092488; OQ102487-OQ102493; OQ257005-OQ257011; OQ271213-OQ271221; OQ540726-OQ540738; OQ553798; OQ573689-OQ573698; OQ652110-OQ652116; OQ675828-OQ675832; OQ866612, OQ866613, OQ866615, OQ866617, OQ866619, OQ866621, OQ866623.

## 3. Results

### 3.1. Sampling

In this study, ticks were collected in two sites (Om-Bo and Om-Zn) of Omsk province, Western Siberia. The investigation included 145 ticks collected from small mammals in the site Om-Bo and examined without preliminary molting. In addition, 20 ticks from the site Om-Bo and 115 ticks from the site Om-Zn collected from mammals and molted under laboratory conditions were tested (Table 2). As *Rickettsia* spp. is efficiently transmitted transovarially, the study of non-molted ticks of all stages can be informative.

Of 145 non-molted ticks taken from 29 rodents in site Om-Bo, 62 *I. apronophorus*, 59 *I. persulcatus* and 24 *I. trianguliceps* ticks were identified using molecular methods. Among larvae and nymphs collected from mammals in site Om-Bo and molted under laboratory conditions, five *I. apronophorus*, 14 *I. persulcatus* and one *I. trianguliceps* were determined. Among ticks collected in site Om-Zn and molted in the laboratory, there were five *I. apronophorus*, 87 *I. persulcatus* and 23 *I. trianguliceps* ticks.

### 3.2. Detection of Rickettsia spp. in Non-Molted Ticks

*Ixodes* spp. larvae, nymphs and adults collected from mammals were examined for the presence of rickettsial agents. *Rickettsia* spp. DNA was detected in 74.2% (46/62) non-molted *I. apronophorus*, 83.1% (49/59) *I. persulcatus* and 66.7% (16/24) *I. trianguliceps* ticks (Table 2). Among infected *I. apronophorus*, 44 ticks were infected with *R. helvetica*, and single ticks carried DNA of “*Ca.* R. tarasevichiae” or both *R. helvetica* and “*Ca*. R. tarasevichiae”. This was the first finding of rickettsiae in *I. apronophorus*. Forty-eight *I. persulcatus* carried DNA of “*Ca*. R. tarasevichiae”, and one tick contained DNA of both “*Ca*. R. tarasevichiae” and *R. raoultii*. As for *I. trianguliceps* ticks, *R. helvetica* was identified in nine larvae, while “*Ca*. R. uralica” was determined in all infected nymphs and adults (*n* = 6) and one larva (Table 2).

### 3.3. Detection of Rickettsia spp. in Molted Ixodes spp. Ticks

Among molted *Ixodes* spp., the prevalence of various rickettsiae substantially varied between different tick species (Table 2). Molted *I. apronophorus* ticks from both sites were infected only with *R. helvetica* (3/5 ticks in site Om-Bo and 4/5 ticks in site Om-Zn), whereas “*Ca*. R. uralica” (3/24; 12.5%) and “*Ca.* R. tarasevichiae” (2/24; 8.3%) were found in molted *I. trianguliceps* (Table 2). Molted *I. persulcatus* ticks from both sites were most frequently infected with “*Ca.* R. tarasevichiae” (57.1% and 85.1% ticks in sites Om-Bo and Om-Zn, respectively). *Rickettsia helvetica*, “*Ca*. R. uralica”, and a new *Rickettsia* genetic variant, *Rickettsia* sp. Om-113/4_Iper_m, were identified only in single *I. persulcatus*. In this study, “*Ca*. R. uralica” was revealed for the first time in a molted *I. persulcatus* tick (Table 2).

Thus, in molted ticks, “*Ca*. R. tarasevichiae” was detected significantly more often in *I. persulcatus*, *R. helvetica*—in *I. apronophorus*, and “*Ca.* R. uralica”—in *I. trianguliceps* compared to other tick species (*p* < 0.001 in all cases).

### 3.4. Genotyping of Rickettsia raoultii and Candidate Species

“*Candidatus* R. tarasevichiae” was identified using species-specific PCR, whereas other rickettsial species were determined by *glt*A gene sequencing. All “*Ca.* R. tarasevichiae” isolates from *I. trianguliceps* and *I. apronophorus* and 20 “*Ca*. R. tarasevichiae” isolates from *I. persulcatus* were genetically characterized by the *glt*A gene. In addition, fragments of the 16S rRNA (732 bp), *glt*A (684 bp), *omp*A (omp AI, 506 bp and omp AIV, 776 bp), *omp*B (864 bp), *sca*4 (1185 bp), *htr*A (499 bp) genes, groESL operon (1519 bp), and 23S-5S IGS region (334 bp) were sequenced for “*Ca.* R. tarasevichiae” isolates from non-molted and molted *I. persulcatus* and molted *I. trianguliceps* (Appendix A). The determined sequences of “*Ca*. R. tarasevichiae” from different tick species were identical and corresponded to those previously identified in *I. persulcatus* ticks from the Russian Far East (OP603104, OP612303-OP612310).

“*Candidatus* R. uralica” isolates were genetically characterized by sequencing of the *glt*A (790 bp), *omp*A (384 bp), *omp*B (1274 bp), and *sca*4 (786 bp) gene fragments. The obtained sequences showed 100% identity between themselves and the previously determined “*Ca*. R. uralica” sequences (OM293669, OM293671-OM293673). For more detailed genotyping, the sequences of the 16S rRNA (1067 bp), *glt*A (1022 bp), *omp*A (two fragments: 578 bp and 3116 bp), *omp*B (4871), *sca*4 (2971 bp), *htr*A (497 bp) genes, groESL operon (1481 bp) and 23S-5S IGS (354 bp) were determined for four “*Ca*. R. uralica” samples from *I. trianguliceps*, including one molted tick (Appendix A). The obtained sequences of each genetic locus showed 100% identity.

*Rickettsia raoultii* and a new rickettsial genetic variant were revealed in single ticks; they were genotyped only by *glt*A gene. *Rickettsia raoultii* isolate from an *I. persulcatus* larva (726 bp) differed by one mismatch from *R. raoultii* RpA4 genotype (DQ365803). A new genetic variant *Rickettsia* sp. Om-113/4_Iper_m from a molted *I. persulcatus* (373 bp) differed by two nucleotide substitutions from *R. heilongjiangensis* and *R. slovaca* (CP002912 and U59725, respectively).

### 3.5. Genotyping of R. helvetica

All *R. helvetica* isolates were genetically characterized by sequencing *glt*A fragments with 840 bp length, and five sequence variants were found. For a subset of specimens with various *glt*A sequences, the *omp*B (1255 bp), *sca*4 (783 bp), and 16S rRNA (684 bp) gene fragments were additionally sequenced. A comparison of the obtained sequences showed the presence of six sequence variants, which varied by 2–8 substitutions. All obtained sequences differed from those of prototype *R. helvetica* strain C9P9 (AICO01000001) and isolates from the Russian Far East (OQ209952, OQ209953, OQ257004) (Figure 2A).

Phylogenetic analysis based on *glt*A-*omp*B-*sca*4 concatenated sequences demonstrated that the obtained sequences belong to three genetic lineages I–III (Figure 3). Specimens from lineage I (European lineage) clustered together with *R. helvetica* str. C9P9, a prototype *R. helvetica* strain isolated from *I. ricinus* from Switzerland. Studied specimens differed from the C9P9 strain by single substitutions in the *omp*B or *glt*A genes (Figure 2A). Sequences from this lineage were identified in site Om-Zn in three molted *I. apronophorus* and one molted *I. persulcatus* (Table 3). Sequences from lineage II (*I. trianguliceps* lineage) were identical to those previously found in two feeding *I. trianguliceps* nymphs from Omsk Province (GenBank KR150775, KR150777, KR150781, KR150786) [14]. In this study, nine *I. trianguliceps* larvae and one molted *I. apronophorus* from site Om-Bo contained DNA of *R. helvetica* from lineage II (Table 3 and Appendix A). Lineage III (*I. apronophorus* lineage) was the most abundant and contained only novel sequences that were determined in 48 *I. apronophorus*, mainly from the site Om-Bo (Table 3 and Appendix A). This lineage was genetically diverse; the sequence of a specimen Om-103_Iapr differed from others by one substitution in the *glt*A gene, whereas sequences of five specimens differed by one substitution in the highly conserved 16S rRNA gene (Figure 2A). Notably, all sequences with a unique substitution in the 16S rRNA gene were found in larvae collected from vole 79. In addition, some other sequences previously detected in *I. persulcatus* from the Far East formed lineage IV (the Far Eastern lineage) on the constructed phylogenetic tree (Figure 3).

Since many *R. helvetica* isolates from the GenBank database were characterized by *omp*B gene, we used this genetic locus to analyze available *R. helvetica* sequences from other regions. The phylogenetic tree, which was reconstructed using the *omp*B gene fragment with a length of 2684 bp, demonstrated the presence of the same four well-supported genetic lineages I–IV (Figure 4). As a result of phylogenetic analysis, genetic lineage I was supplemented with *R. helvetica* specimens from *I. ricinus* from Germany (MF163037, HQ232244-HQ232251) and *I. persulcatus* from Western Siberia (Novosibirsk province) (KU310591) (Figure 4) Lineage II additionally included 32 *R. helvetica* specimens identified in *I. persulcatus* from Komi Republic [31]. Sequences from the Komi Republic differed from the studied *R. helvetica* sequences from *I. trianguliceps* by one substitution in each of the *omp*B and *glt*A genes (Figure 2B and Figure 4). As for genetic lineage IV, sequences from *I. persulcatus* from the continental part of the Far East (KT825966, KT825970) also corresponded to this lineage in addition to those from Sakhalin and Putyatin islands [15].

For more detailed genotyping, sequences of the 16S rRNA (1070 bp), *glt*A (1037 bp), *omp*A (1417 bp), *omp*B (3100 bp), *sca*4 (2398 bp) and *htr*A (499 bp) genes, as well as 23S-5S IGS region (489 bp) and groESL operon (1528 bp), were determined for five *R. helvetica* isolates, belonging to different lineages. Notably, sequences of the groESL operon were identical to all known *R. helvetica* samples and these sequences were not used for phylogenetic analysis. All other examined genetic loci have polymorphic sites, with the *omp*B gene being the most variable. Among coding sequences, nucleotide substitutions in 15/25 polymorphic sites were non-synonymous (Figure 2B). The obtained concatenated sequence of isolate Om-74_Iapr_m from lineage I differed from the sequence of *R. helvetica* str. C9P9 by one substitution in the *omp*B gene (Figure 2B). Sequences of the specimens from lineages II and III varied between themselves by 20 substitutions and differed from the sequences of *R. helvetica* str. C9P9 and Far Eastern isolate Skh-7_Iper (OQ209950-OQ209956, OQ257004) by 12–16 substitutions (Figure 2B). The comparison of polymorphic sites from different genetic loci showed that *omp*B and *sca*4 gene fragments could be used to reliably differentiate specimens from various genetic lineages (Figure 2B). Notably, phylogenetic analysis based on the *sca*4 gene demonstrated that *R. helvetica* isolate from *I. persulcatus* from Japan (FJ358501) [27] can also be referred to as the Far Eastern lineage.

Phylogenetic tree based on 16S-*glt*A-*omp*A-*omp*B-*sca*4-*htr*A-IGS concatenated sequences (9779 bp) (Figure 5) showed the presence of four well-supported genetic lineages, which correspond to those that were identified based on analysis of shorter *glt*A-*omp*B-*sca*4 concatenated sequences and the *omp*B gene fragment with a length of 2684 bp (Figure 3 and Figure 4).

## 4. Discussion

In Russian Siberia, several *Rickettsia* species are abundant. Different *Rickettsia* species are usually associated with certain tick species. Thus, “*Ca*. R. tarasevichiae”, “*Ca.* R. uralica” and *R. raoultii* are closely associated with *I. persulcatus*, *I. trianguliceps* and *Dermacentor* spp., respectively [11,14,16,32,33,34]. As for *R. helvetica*, this agent is associated with *Ixodes* spp. and is prevalent in *I. ricinus* in Europe and *I. persulcatus* from Sakhalin Island (the Far East) and Komi Republic (European part of Russia) [15,19,31,35].

The study of rickettsiae agents in ticks from sympatric areas is of particular interest because it makes it possible to compare pathogen-tick association for different tick species from the same location. This study includes *I. apronophorus*, *I. persulcatus* and *I. trianguliceps* ticks collected in two sites in the Omsk Province. In site Om-Bo, the abundance of all these tick species was high, whereas in site Om-Zn, *I. persulcatus* dominated and the prevalence of *I. apronophorus* was low [8].

In this study, *Rickettsia* spp. were first found in *I. apronophorus*. *Rickettsia helvetica* was found in 70–80% of molted and non-molted *I. apronophorus* from both locations, indicating a close association of *R. helvetica* with *I. apronophorus*. In addition to *R. helvetica*, “*Ca.* R. tarasevichiae” was identified in 3% of non-molted *I. apronophorus* (Table 2).

Expectedly, “*Ca.* R. tarasevichiae” prevailed in *I. persulcatus*, occurring in more than 80% of molted and non-molted ticks. Other *Rickettsia* spp. were found in *I. persulcatus* only in single cases. Previously, a similarly high prevalence of “*Ca*. R. tarasevichiae” in questing adult *I. persulcatus* has been observed in various regions of the Asian part of Russia in Omsk Province, but not in *I. persulcatus* from Sakhalin Island, Komi Republic, and Estonia [11,14,15,16,31,32]. Surprisingly, one molted *I. persulcatus* was infected with “*Ca*. R. uralica”, despite “*Ca*. R. uralica” was not found in any of the over 500 previously analyzed questing *I. persulcatus* from Omsk province [14].

As for *I. trianguliceps*, three *Rickettsia* species, “*Ca.* R. uralica”, “*Ca*. R. tarasevichiae” and *R. helvetica*, were found in this tick species. Notably, the prevalence of *Rickettsia* spp. substantially varied depending on the sampling site and developmental stage of *I. trianguliceps* (Table 2). Association of “*Ca.* R. uralica” with *I. trianguliceps* has been previously recorded in other locations of Omsk province and Estonia [14,17], whereas “*Candidatus* R. thierseensis” (a genetic variant of “*Ca.* R. uralica”) was found in one *I. ricinus* in Austria [24,36]. The findings of “*Ca.* R. uralica” in human-biting *I. persulcatus* and *I. ricinus* may indicate the potential threat of this rickettsial species to humans.

“*Candidatus* R. tarasevichiae”, recently recognized as a pathogenic species [37,38,39,40], is reliably associated with *I. persulcatus*; however, in rare cases, it has been found in other tick species, namely *I. pavlovskyi*, *Dermacentor* spp., and *Haemaphysalis* spp. [11,16,22,41,42]. In this study, “*Ca*. R. tarasevichiae” was found in two molted *I. trianguliceps* in the Om-Zn site (Table 2), which is consistent with the previous detection of “*Ca*. R. tarasevichiae” in feeding *I. trianguliceps* nymphs and adults from Omsk province [14]. It is noteworthy that *Rickettsia* spp. were first identified in molted *I. trianguliceps* ticks, which demonstrates their ability to transmit transstadially both “*Ca.* R. uralica” and “*Ca.* R. tarasevichiae” (Table 2).

Surprisingly, *R. helvetica* was found only in *I. trianguliceps* larvae but not in nymphs and adults (Table 2 and Appendix A). As all *R. helvetica*-infected larvae were collected only from two voles, the observed discrepancy may be explained by the insufficient number of *I. trianguliceps* studied and the uneven distribution of infected and uninfected larval offspring from different females. This uneven distribution of larvae also explains the fact that all *R. helvetica* specimens with a unique substitution in the 16S rRNA gene were identified in *I. apronophorus* larvae (but not adults) collected from the same vole.

A number of “*Ca.* R. uralica”, “*Ca.* R. tarasevichiae”, and *R. helvetica* isolates were genetically characterized by sequencing nine genetic loci. “*Candidatus* R. uralica” and “*Ca*. R. tarasevichiae” isolates were shown to be highly conserved; 100% identity was shown for the obtained sequences of “*Ca*. R. uralica” specimens from *I. persulcatus* and *I. trianguliceps* and for “*Ca.* R. tarasevichiae” isolates from Omsk province (this study) and the Russian Far East [43]. 

On the contrary, analyzed *R. helvetica* isolates were diverse. Six sequence variants of *R. helvetica* belonging to three genetic lineages were identified. Only *R. helvetica* specimens from lineage II were found in infected *I. trianguliceps*, whereas *R. helvetica* from lineages I–III were identified in *I. apronophorus* (Table 3). The observed high divergence may be related to the wide range of *R. helvetica* carriers identified in this and previous studies: *I. ricinus*, *I. pavlovskyi*, *I. persulcatus*, *I. apronophorus*, *I. trianguliceps*, *I. hexagonus*, *I. arboricola*, *I. ovatus* and *I. monospinosus* [3,11,15,31,35,44].

To date, there is no reliable data confirming *Rickettsia* spp. co-feeding transmission [45]. Although this transmission may occur in artificial conditions (in the case of *R. rickettsia*), its impact on pathogen transmission in nature seems insignificant [46]. Our study of non-molted larvae indicated that at least *R. helvetica* and “*Ca.* R. tarasevichiae” cannot be effectively transmitted between different *Ixodes* species as a result of simultaneous feeding on small mammals. Indeed, with single exceptions, *I. persulcatus* larvae were infected with “*Ca.* R. tarasevichiae”, *I. apronophorus*—with *R. helvetica* from lineage III, and *I. trianguliceps*—with *R. helvetica* from lineage II (Table 2, Table 3 and Appendix A). Notably, the association between tick species and rickettsial species/sequence variant was retained when larvae of different species were fed on the same animal (Appendix A: rodents BU75, 79, 158). However, it cannot be ruled out that single findings of “*Ca.* R. tarasevichiae” and “*Ca.* R. uralica” in atypical tick carriers may be due to rare cases of co-feeding transmission.

The long-length sequences of *R. helvetica* (above 11,500 bp) were determined for specimens belonging to various phylogenetic groups. Analysis of *R. helvetica* sequences from this study and available sequences from Europe (mainly from Germany) [47], North Western Russia [31], Western Siberia, and the Far East [15,27] demonstrated a high genetic variability of *R. helvetica*. The analyzed sequences can be reliably assigned to four genetic lineages. However, the association of different lineages with specific tick species and territories was not observed in all cases. Thus, although the European genetic lineage (lineage I) dominated *I. ricinus* in Europe, it was also found in *I. persulcatus* and *I. apronophorus* from Western Siberia. Similarly, *I. trianguliceps* genetic lineage (lineage II) was found in both *I. persulcatus* and *I. trianguliceps* ticks from two remote regions of Russia. On the contrary, *I. apronophorus* genetic lineage (lineage III) was identified only in *I. apronophorus* from Western Siberia; the Far Eastern genetic lineage (lineage IV) was identified only in *I. persulcatus* from the Far East.

It has been shown that *R. helvetica* can cause rickettsiosis in humans, mainly associated with fever, headache, and myalgias [3,44,48]. However, data on the genetic variability of *R. helvetica* is limited by a small number of studied geographic regions and has been somewhat extended as a result of this study. Further genetic characterization of *R. helvetica* isolates from other regions and/or other tick species is required to assess the prevalence and distribution of different genetic lineages of *R. helvetica* and to fill a gap in our knowledge of *R. helvetica* biodiversity. It cannot be ruled out that different genetic lineages of *R. helvetica* may differ in their pathogenic properties.

## Figures and Tables

**Figure 1 microorganisms-11-01215-f001:**
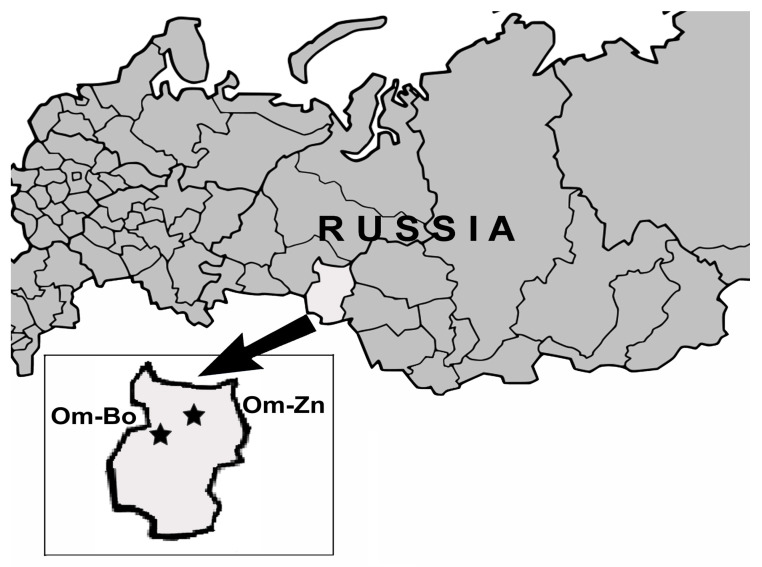
Sites of specimen collection in Omsk province of Western Siberia. Stars shows the locations of sampling sites.

**Figure 2 microorganisms-11-01215-f002:**
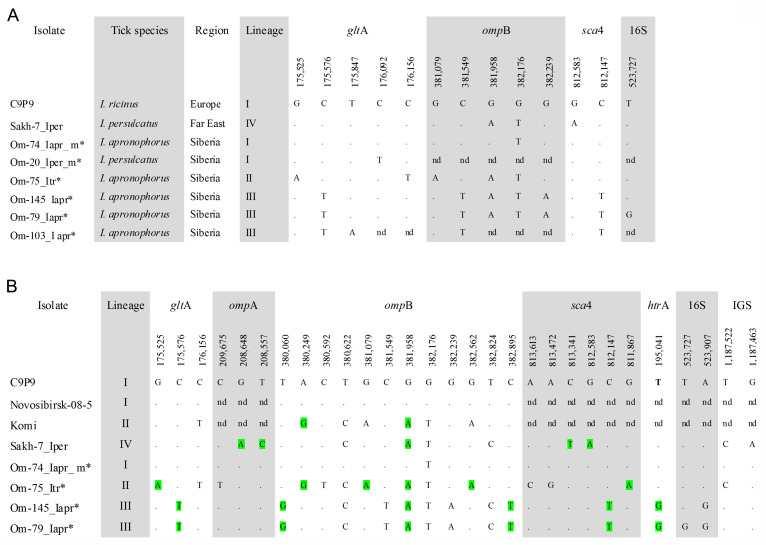
(**A**) Condensed alignment of *glt*A (840 bp), *omp*B (1255 bp), *sca*4 (783 bp), and 16S rRNA (684 bp) gene fragments of *R. helvetica* sequence variants from this study; (**B**) Condensed alignment of *glt*A (1037 bp), *omp*A (1417 bp), *omp*B (3100 bp), *sca*4 (2398 bp), *htr*A (499 bp), 16S rRNA (1070 bp) genes and 23S-5S IGS region (489 bp) of *R. helvetica* genetic lineages. Variable nucleotide positions are given according to sequence *R. helvetica* strain C9P9 (AICO01000000). * the specimens from this study. Non-synonymous polymorphic sites are highlighted in green.

**Figure 3 microorganisms-11-01215-f003:**
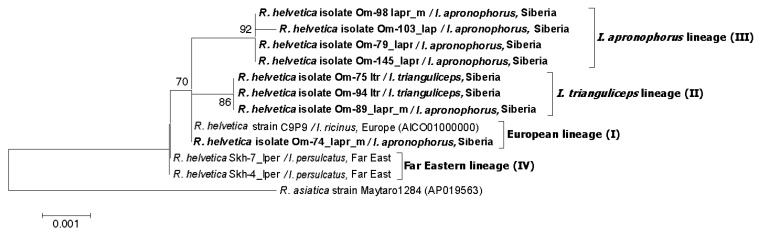
The phylogenetic tree was constructed using the ML method. Three gene fragments were concatenated (*glt*A-*omp*B-*sca*4), and a total of 2259 positions were analyzed. The scale bar indicates an evolutionary distance of 0.001 nucleotide per position in the sequence. Significant bootstrapping values (>70%) are shown on the nodes. The sequences of *R. helvetica* determined in this study are in boldface.

**Figure 4 microorganisms-11-01215-f004:**
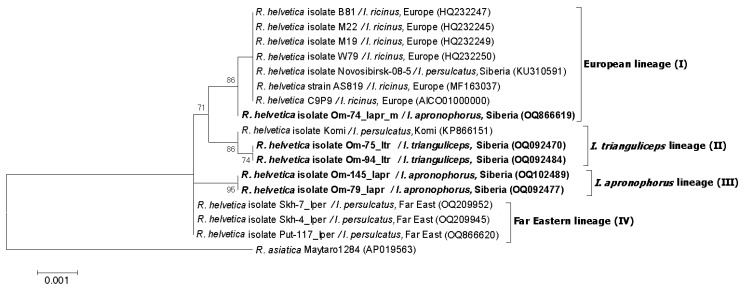
The phylogenetic tree constructed by the ML method based on nucleotide sequences of the 2684 bp fragment of the *ompB* gene of *R. helvetica*. The sequence of *R. asiatica* was used as outgroup. The scale bar indicates an evolutionary distance of 0.001 nucleotide per position in the sequence. Significant bootstrapping values (>70%) are shown on the nodes. The sequences of *R. helvetica* determined in this study are in boldface.

**Figure 5 microorganisms-11-01215-f005:**
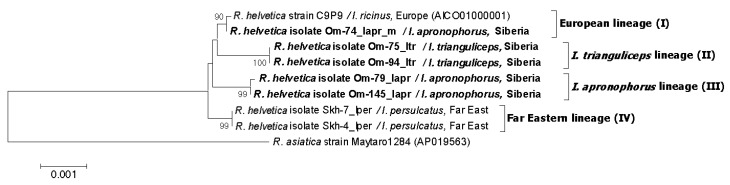
The phylogenetic tree was constructed using the ML method. Seven loci were concatenated (*glt*A-*omp*A-*omp*B-*sca*4-*htr*A-16SrRNA-IGS), and a total of 9779 positions were analyzed. The sequence of *R. asiatica* was used as outgroup. The scale bar indicates an evolutionary distance of 0.001 nucleotide per position in the sequence. Significant bootstrapping values (>70%) are shown on the nodes. The sequences of *R. helvetica* determined in this study are in boldface.

**Table 2 microorganisms-11-01215-t002:** Detection of *Rickettsia* spp. in molted and non-molted ticks.

Site/Tested Ticks	Tick Species	Tick Stage	No. of Ticks	No. (%) of Ticks Containing DNA of				
				All Rickettsiae	R.helv	R.tar	R.ural	Mixed Infection
Om-Bo/	I.apr	Larvae	47	34	33	1	0	0
non-molted		Nymphs	5	4	4	0	0	0
ticks		Females	9	7	6	0	0	1/R.tar + R.helv
		Males	1	1	1	0	0	0
		**All stages**	**62**	**46 (74.2)**	**44 (71.0)**	**1 (1.6)**	**0**	**1 (1.6)**
	I.pers	Larvae	55	46	0	45	0	1/R.tar + R.raol
		Nymphs	4	3	0	3	0	0
		**All stages**	**59**	**49 (83.1)**	**0**	**48 (81.4)**	**0**	**1 (1.7)**
	I.tr	Larvae	17	10	9	0	1	0
		Nymphs	4	4	0	0	4	0
		Females	3	2	0	0	2	0
		**All stages**	**24**	**16 (66.7)**	**9 (37.5)**	**0**	**7 (29.2)**	**0**
Om-Bo/	I.apr	Nymphs	2	1	1	0	0	0
molted		Females	1	1	1	0	0	0
ticks		Males	2	1	1	0	0	0
		**All stages**	**5**	**3 (60)**	**3 (60)**	**0**	**0**	**0**
	I.pers	Nymphs	3	3	0	2	1	0
		Females	7	5	0	5	0	0
		Males	4	1	0	1	0	0
		**All stages**	**14**	**9 (64.3)**	**0**	**8 (57.1)**	**1 (7.1)**	**0**
	I.tr	Nymphs	1	1	0	0	1	0
		**All stages**	**1**	**1**	**0**	**0**	**1**	**0**
Om-Zn/	I.apr	Nymphs	4	3	3	0	0	0
molted		Males	1	1	1	0	0	0
ticks		**All stages**	**5**	**4 (80)**	**4 (80)**	**0**	**0**	**0**
	I.pers	Nymphs	23	23	0	22	0	1/R.tar + R.sp
		Females	36	29	0	28	0	1/R.tar + R.helv
		Males	28	22	0	22	0	0
		**All stages**	**87**	**74**	**0**	**72 (82.8)**	**0**	**2 (2.3)**
	I.tr	Nymphs	10	0	0	0	0	0
		Females	5	2	0	1	1	0
		Males	8	2	0	1	1	0
		**All stages**	**23**	**4 (17.4)**	**0**	**2 (8.7)**	**2 (8.7)**	**0**
**Total**	**I.apr**	**All stages**	**72**	**53 (73.6)**	**51 (70.8)**	**1 (1.4)**	**0**	**1 (1.4)**
	**I.pers**	**All stages**	**160**	**132 (82.5)**	**0**	**128 (80.0)**	**1 (0.6)**	**3 (1.9)**
	**I.tr**	**All stages**	**48**	**21 (43.8)**	**9 (18.8)**	**2 (4.2)**	**10 (20.8)**	**0**

Abbreviations: I.apr—*I. apronophorus*; I.pers*—I. persulcatus*; I.tr—*I. trianguliceps*; R.helv—*R.helvetica*; R.ta—“*Ca.* R. tarasevichiae”; R.ural—“*Ca*. R. uralica”; R.raol—*R. raoultii*, R.sp—*Rickettsia* sp. Om-113/4_Iper_m.

**Table 3 microorganisms-11-01215-t003:** Prevalence of different sequence variants of *R. helvetica* in *Ixodes* spp.

Sites	Tick Species	Tested Ticks	No of Tested Ticks	No of *R. helvetica* Positive Ticks	No of *R. helvetica* Samples Belonging to		
					Lineage I	Lineage II	Lineage III
Om-Bo	*I. apronophorus*	non-molted	62	45	0	0	45
		molted	5	3	0	1	2
		**subtotal**	**67**	**48**	**0**	**1**	**47**
	*I. persulcatus*	non-molted	59	0	0	0	0
		molted	14	0	0	0	0
		**subtotal**	**73**	**0**	**0**	**0**	**0**
	*I. trianguliceps*	non-molted	24	9	0	9	0
		molted	1	0	0	0	0
		**subtotal**	**25**	**9**	**0**	**9**	**0**
Om-Zn	*I. apronophorus*	molted	5	4	3	0	1
	*I. persulcatus*	molted	87	1	1	0	0
	*I. trianguliceps*	molted	23	0	0	0	0
**Both**	** *I. apronophorus* **	**total**	**72**	**52**	**3**	**1**	**48**
**sites**	** *I. persulcatus* **	**total**	**160**	**1**	**1**	**0**	**0**
	** *I. trianguliceps* **	**total**	**48**	**9**	**0**	**9**	**0**

## Data Availability

The data presented in this study are available in Appendix A.

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
