# Peer review of "Novel Genetic Lineages of Rickettsia helvetica Associated with Ixodes apronophorus and Ixodes trianguliceps Ticks"

_microorganisms, 2023, doi:10.3390/microorganisms11051215_

Round 1

Reviewer 1 Report

The manuscript titled “Novel genetic lineages of Rickettsia helvetica associated with Ixodes apronophorus and Ixodes trianguliceps ticks” reported the prevalence and genetic diversity of Rickettsia spp. in I. apronophorus, I. persulcatus, and I. trianguliceps ticks from their sympatric habitats in Western Siberia, Russian. The study reported some novel findings. The experiments were overall solid and the manuscript was well presented.

Minor points:

First page: Lines 12-13 “..in Ixodes apronophorus, Ixodes persulcatus, and Ixodes trianguliceps ticks”, here Ixodes can be safely written as “I.” , be type in italics.

Table 1: To be consistent for the gene names, “16S rRNA” may be also written in italics.

Figure 4: Bootstrap numbers may be provided in the legend.

Author Response

Point 1: First page: Lines 12-13 “..in Ixodes apronophorusIxodes persulcatusand Ixodes trianguliceps ticks”, here Ixodes can be safely written as “I.” , be type in italics.

Response 1: We have retained the full genus name, because the scientific name should be written in full at the first mention.

Point 2: Table 1: To be consistent for the gene names, “16S rRNA” may be also written in italics.

Response 2: Corrected.

Point 2: Figure 4: Bootstrap numbers may be provided in the legend.

Response 3: Unfortunately, we did not understand this remark. The bootstrap numbers should be provided in the nodes of phylogenetic trees. It was indicated in the legends that "Significant bootstrapping values (>70%) are shown on the nodes".

Reviewer 2 Report

Dear authors,

Here are comments and suggestions:

- There are many Keywords, some of which are repeated in the title of the manuscript. Reconsider the terms. Suggestion: Ixodes; “Candidatus Rickettsia uralica”; “Candidatus Rickettsia tarasevichiae”; genetic lineages; phylogenetic analysis.

- Line 31: Replace "Ixididae" with "Ixodidae".

- Line 35: In each species cited from Ixodes for the first time, do not abbreviate the genus name.

- Line 49: Replace "Borrelia burgdorferi sensu lato" with "B. burgdorferi s.l.".

- Line 95: The format "+10-15°C" is confuse. I suggest removing the plus symbol, as it will continue to indicate positive temperature.

There are few comments where I congratulate the authors for the complete study presented. An enjoyable and satisfying read. 

Author Response

Point 1: - There are many Keywords, some of which are repeated in the title of the manuscript. Reconsider the terms. Suggestion: Ixodes; “Candidatus Rickettsia uralica”; “Candidatus Rickettsia tarasevichiae”; genetic lineages; phylogenetic analysis.

Response 1: We  have reduced the number of Keywords.

Point 2: - Line 31: Replace "Ixididae" with "Ixodidae".

Response 2: Corrected

Point 3 - Line 35: In each species cited from Ixodes for the first time, do not abbreviate the genus name.

Response 3: Corrected

Point 4 -Line 49: Replace "Borrelia burgdorferi sensu lato" with "B. burgdorferi s.l.".

Response 4: Corrected

Point 5- Line 95: The format "+10-15°C" is confuse. I suggest removing the plus symbol, as it will continue to indicate positive temperature.

Response 5: Corrected